# T-Cell Mediated Immunity in Merkel Cell Carcinoma

**DOI:** 10.3390/cancers14246058

**Published:** 2022-12-09

**Authors:** Kelsey Ouyang, David X. Zheng, George W. Agak

**Affiliations:** 1Division of Dermatology, Department of Medicine, David Geffen School of Medicine at UCLA, Los Angeles, CA 90095, USA; 2Cleveland Clinic Lerner College of Medicine, Case Western Reserve University, Cleveland, OH 44195, USA; 3Department of Dermatology, University Hospitals Cleveland Medical Center, Case Western Reserve University, Cleveland, OH 44106, USA

**Keywords:** Merkel cell carcinoma, T-cell immunity, tumor infiltrating lymphocytes, MCC, MCPyV

## Abstract

**Simple Summary:**

Efforts have been directed toward exploring the pathophysiology underlying virus-positive and virus-negative Merkel cell carcinoma. An increasing amount of research has underlined the role of T-cells in the initiation, progression, and clearance of this rare skin cancer. An updated summary of recent research exploring T-cell-mediated immunity in Merkel cell carcinoma informs future research directions as well as targets for immunotherapy.

**Abstract:**

Merkel cell carcinoma (MCC) is a rare and frequently lethal skin cancer with neuroendocrine characteristics. MCC can originate from either the presence of MCC polyomavirus (MCPyV) DNA or chronic ultraviolet (UV) exposure that can cause DNA mutations. MCC is predominant in sun-exposed regions of the body and can metastasize to regional lymph nodes, liver, lungs, bone, and brain. Older, light-skinned individuals with a history of significant sun exposure are at the highest risk. Previous studies have shown that tumors containing a high number of tumor-infiltrating T-cells have favorable survival, even in the absence of MCPyV DNA, suggesting that MCPyV infection enhances T-cell infiltration. However, other factors may also play a role in the host antitumor response. Herein, we review the impact of tumor infiltrating lymphocytes (TILs), mainly the CD4^+^, CD8^+^, and regulatory T-cell (Tregs) responses on the course of MCC, including their role in initiating MCPyV-specific immune responses. Furthermore, potential research avenues related to T-cell biology in MCC, as well as relevant immunotherapies are discussed.

## 1. Introduction

Merkel cell carcinoma (MCC) is a rare, aggressive neurocutaneous malignancy most often occurring in older White patients [1]. The estimated 5-year recurrence rate is 40–63% and mortality rate is 33–46% [2,3,4]. The incidence of MCC has increased by 95% in the past decade, presumed to be due to an aging population, increased aggregate sun exposure, and higher numbers of immunosuppressed individuals [5]. Similar to other skin cancers, MCC is typically observed on sun-exposed areas such as the head, face, and neck. As suggested above, the risk factors include advanced age, fair skin, ultraviolet (UV) exposure, and a weakened immune system [6]. In particular, recipients of solid organ transplants taking immunosuppressive medications, as well as immunocompromised individuals, are at a higher risk of MCC and have an increased likelihood of recurrence, metastasis, and death [7]. While the exact cause is yet to be elucidated, MCC pathogenesis is typically categorized by two main etiologies: Merkel cell polyomavirus (MCPyV) (80% of cases) or extensive UV exposure (20% of cases) [8] (Figure 1).

Since the emergence of reports suggesting that MCC disproportionately affects immunocompromised individuals, an infectious cause had been suspected. Feng et al. performed the whole transcriptome sequencing of MCC tumors in their search for a pathogenic cause, leading to the discovery that MCPyV DNA was clonally integrated into the genome of MCC tumor cells [9]. This finding underlined the association of MCPyV with MCC [9]. Additional evidence revealing that virus-positive MCC tumors generally possess few somatic mutations suggests that these tumors have MCPyV gene products that may drive MCC tumorigenesis [10,11]. Furthermore, studies on the pathophysiology of virus-positive MCC revealed the functional domains of the small tumor (sT) and large tumor (LT) antigens in mediating MCPyV-associated tumorigenesis [12]. Both antigens are required for MCPyV-positive MCC cell survival and proliferation and have been shown to dysregulate cell pathways by interacting with host proteins, including those regulating tumor suppression, cell cycle progression, and proteasomal activity [13,14,15,16,17].

Although the majority of MCC cases are virally mediated, the true origin of MCC remains debatable [18]. UV radiation (UVR) is a general risk factor for skin cancers, as it can cause both direct skin damage (most frequently through formation of cyclobutane pyrimidine dimers) and mutations (e.g., to p53 tumor suppressor genes), meaning that UVR plays a critical role in tumor initiation [19]. Skin exposure to UVR prompts infiltration by neutrophils, key producers of reactive oxygen species that can further activate downstream signaling pathways associated with cancer-related inflammation [20]. Supporting the association of UVR with the development of MCC are observations that MCC lesions typically occur on sun-exposed areas, primarily in fair-skinned individuals. It has been reported that MCPyV-negative tumors develop as a direct result of UV-associated mutations [21]. Additionally, a dose-dependent increase in mRNA transcripts of the MCPyV sT antigen can be observed upon UVR exposure [10,22,23].

Regardless of the specific etiology, MCPyV and UV-related MCC exhibit clinical and histopathological similarities [24]. Efforts have been directed towards exploring the pathophysiology underlying these two etiologies of MCC. In particular, the exploration of the role T-cells play in the initiation, progression, and clearance of MCC is of interest [25]. In this review, we discuss the role of tumor infiltrating lymphocytes (TILs) in driving the development and clearance of MCC. We begin by addressing the current landscape of knowledge regarding MCC pathogenesis, highlighting how MCPyV infection influences T-cell infiltration and promotes tumor progression. The proposed mechanisms for how T-cells drive the pathogenesis of MCC, specifically regarding initiation of MCPyV-specific immune responses, will be explored. Throughout, we draw connections to specific immune factors within the tumor microenvironment (TME) that relate to the etiology and pathogenesis of MCC. Finally, we will highlight future directions regarding the current immunotherapy paradigm for MCC and potential research avenues related to T-cell biology. By elucidating how T lymphocyte-mediated cellular events relate to initiation, progression, and clearance of MCC, we may consider how T-cell-mediated therapeutic strategies can target the TME to eradicate MCC.

## 2. Tumor Microenvironment in MCC

Aberrant innate and adaptive immune processes contribute to the selection of aggressive clones and the proliferation and metastasis of cancer cells [26]. In the initial stages of tumor development, cytotoxic immune cells, including natural killer (NK) and CD8^+^ T-cells, detect and eliminate immunogenic cancer cells. This tumoricidal attack results in a selection process that favors cells that can evade tumor-specific immune responses [26]. The fate of the tumor largely depends on the balance between antitumor and immune escape processes in the TME—the environment around the tumor consisting of the extracellular matrix and diverse cell types such as stromal, endothelial, and immune cells [27,28,29].

Many biomarkers have been implicated as potential prognosticators in MCC [30,31,32,33,34,35], but the integrity of the host immune system has been argued to be the most significant [36]. Indeed, antigen-specific T-cells detected in patients demonstrate a reduced expression of CD69 and CD25. In addition, 50% of the nonactivated T-cells in MCC express PD-1, a marker of T-cell exhaustion. PD-1 interacts with PD-L1, which is often expressed by MCC cells leading to T-cell dysfunction [37,38]. Previous studies have indicated that the density, location, and organization of immune cells within tumors at the time of diagnosis can reflect the host’s immune response and help predict the course of tumor progression [39,40,41,42]. In various malignancies, including melanoma and colorectal, lung, and breast cancers, increased numbers of immune cells have been associated with improved prognosis [43,44,45,46,47]. While some immune cells, such as cytotoxic T-cells, helper T-cells (T_H_), dendritic cells (DC), and NK cells, provide antitumor responses, others, such as regulatory T-cells (Tregs), have been associated with immunosuppressive functions favoring tumor progression [48]. These opposing effects are further mediated by the production of chemokines and cytokines that can facilitate bidirectional crosstalk between cancer cells and the immune microenvironment [49,50]. Further efforts aimed at quantifying tumor immune cells and characterizing the immune landscape in MCC may help identify individuals at an increased risk of MCC recurrence or mortality.

## 3. Role of Tumor Infiltrating Lymphocytes in MCC

Within most tumors exist TILs, a heterogeneous lymphocyte population composed primarily of T-cells that penetrates the tumor upon the recognition of cancer antigens, therefore potentially contributing to tumor destruction or escape [51]. The prognostic significance of TILs in MCC has been recognized for many years [38,52]. The immunostaining of MCC tumor cryosections has revealed tumor infiltration by various subpopulations of T-cells, including effector, regulatory, and central memory (T_CM_) [38] (Figure 2). MCCs with higher levels of T-cells expressing CLA (cutaneous lymphocyte-associated antigen) demonstrate increased infiltration, while the subset of MCC tumors with T-cells lacking CLA expression fail to penetrate the tumor. Although this CLA-negative peritumoral pattern is typically associated with poor survival, the presence of CLA-positive TILs alone was not necessarily protective, as these individuals still experience MCC recurrence and mortality [38]. Among T-cells that infiltrate MCC, researchers have observed decreased CD69 expression compared to T-cells from normal, non-inflamed skin, suggesting that the MCC:TME modulates the expression of T-cell-associated activation markers [38]. Observational studies using data from the National Cancer Database have found the presence of TILs to be associated with improved overall survival in MCC [53,54]. Although researchers recognize that TILs may portend improved prognosis in MCC, it will be important to tease apart the differential clinical relevance of individual subsets of TILs in order to inform management (e.g., optimizing the use of adjuvant immunotherapeutic agents to enhance antitumor immune response) of MCC.

### 3.1. CD8^+^ and CD4^+^ T-Cells

Abundant numbers of infiltrating CD4^+^ and CD8^+^ T-cells have been observed in MCC tumors. The high numbers of intra-tumoral CD8^+^ T-cells infiltrating the MCC microenvironment have been associated with MCC survival, regardless of the tumor stage at diagnosis [55,56]. In MCC, the MCPyV gains oncogenic potential via rare integration and T-antigen (T-Ag) truncation mutations [57]. The association of high CD8^+^ T-cell counts with improved survival is mediated by virus-specific CD8^+^ T-cells recognizing a broad range of peptides and oncoproteins expressed by tumor cells to effect cell lysis [25]. The mutations within these epitopes may exert their influence on TIL recognition via allostery and alterations in local folding, leading to reduced CD8-mediated cytotoxicity. As demonstrated by Samimi et al., CD8^+^ T-cells may additionally have antitumor activity in virus-negative MCC through their recognition of the melanoma-associated antigen 3, which is often expressed in MCC [58].

Immuno-oncology studies have traditionally focused on the role of CD8^+^ T-cells in antitumor immune responses, largely due to the ~100–1000-fold lower relative frequency of antigen-specific CD4^+^ T-cells compared to CD8^+^ T-cells [59]. However, the fact that MCC results from MCPyV oncoproteins in 80% of cases has facilitated the study of the role of tumor-specific CD4^+^ T-cell help in MCC antitumor activity. While most CD4^+^ T-cells act as helper T-cells that activate CD8^+^ T-cells, some also activate B-cells to differentiate into plasma cells or memory B-cells, as evidenced by the high prevalence of MCPyV-specific antibodies in healthy individuals [25]. MCPyV antibodies have been detected in a majority (77%) of the general population without observed variations according to sex or age [60]. Furthermore, the administration of MCPyV viral-like proteins (VLPs) induced a strong immune response in mice, suggesting a good requisite for the development of preventive VLP-based cancer vaccines, since VLPs can trigger a strong CD4^+^ T-cell-mediated immune responses against oncoviruses [60,61]. Furthermore, Longino et al. identified CD4^+^ T-cell responses against six MCPyV epitopes, most notably the WED_LT209-228_ epitope within a key oncogenic region of the MCPyV LT antigen [62]. This epitope was found to include a conserved, essential oncogenic domain that upon modification disabled the binding of MCPyV to the retinoblastoma tumor suppressor gene [57,62]. CD4^+^ T-cells specific to the WED_LT209-228_ epitope were enriched 250-fold within MCC tumor samples compared to blood and exhibited diverse T-cell receptor (TCR) repertoires, therefore suggesting in vivo antigen recognition and therapeutic potential of CD4^+^ T-cell responses targeting of specific cancers [62].

### 3.2. Regulatory T-Cells

The higher proportion of Tregs within the TME may promote tumor development and proliferation [63]. Tregs secrete anti-inflammatory cytokines, such as IL-10, transforming growth factor (TGF)-β, and IL-35, and express inhibitory receptors, lymphocyte-activation gene 3 (LAG-3), and T-cell immunoreceptor with Ig and ITIM domain (TIGIT) that suppress DC function and T_H_1 responses [38,64,65,66]. TIGIT is expressed in immune cells, including T-cells, natural killer cells, and Tregs. This coinhibitory molecule is known to bind to two ligands CD155 and CD112 expressed by tumor cells and APCs within the TME. It has been demonstrated that TIGIT^+^ Tregs are highly suppressive and enriched in cutaneous tumors, such as melanoma. Therefore, a dual PD-1/TIGIT blockade is a potential therapy for cancers, including MCC [67,68,69].

Dowlatshahi et al. observed that MCC TILs comprised primarily CD25^+^ FOXP3^+^ Tregs, which contrasts with normal human skin that contains a higher proportion of CD25^+^ FOXP3^−^ activated T-cells [38]. Additionally, MCC tumors containing CD8^+^ FOXP3^+^ Tregs contained plasmacytoid dendritic cells (pDCs), which are known to secrete cytokines that promote the formation of CD8^+^ Tregs [38]. Together, CD8^+^ FOXP3^+^ Tregs and pDCs contribute to an immunosuppressive TME, which has been associated with resistance to immunotherapy and poor patient prognosis [70,71] (Figure 2).

### 3.3. Other T-Cell Subsets

Gherardin et al. found a large proportion of T-cells in MCC that expressed neither CD4 nor CD8 on their surfaces, termed double negative (DN) T-cells [52]. Using flow cytometry, they revealed that γδ T-cells represent the majority of DN T-cells in MCC tumors. T-cells have classically been divided into αβ or γδ lineages based on the expression of either the αβ or γδ T-cell receptor (TCR) [52,72]. Unlike the αβ T-cell lineage, γδ T-cells have non-major histocompatibility complex (MHC) restricted antigen recognition, which allow them to mediate quick immune inflammatory responses [73]. Following activation, γδ T-cells rapidly proliferate and produce inflammatory cytokines, such as IFN-γ and IL-17, as well as release cytotoxic granules such as granzymes and perforin [74]. γδ T-cells have been shown to be able to exert direct tumor killing effects, as well as regulate other proinflammatory and antigen-driven responses that may involve other immune cells [73]. The abundance of γδ T-cells have been correlated with improved survival in numerous cancers, including MCC [52]. More studies are required to elucidate the role of γδ T-cells in immune surveillance, regulation and antitumor immunity.

## 4. T-Cell Receptor Repertoires

The sequencing of TCR repertoires has been used to study various medical conditions, such as autoimmune diseases, infections, and cancer [75,76,77]. In recent years, researchers have sought to associate the TCR repertoire with MCC patients’ risk for disease-specific morbidities and how they might respond to specific immunotherapies. Farah et al. evaluated the TCR repertoire associated with 72 primary MCCs and correlated metrics of the TCR repertoire with clinicopathologic characteristics and patient outcomes [36]. They found that higher TCR diversity was associated with fewer metastases, lower stages at presentation, and longer times until first lymph node metastasis [36]. In a separate study, Spassova et al. observed that a predominance of T_CM_ with high TCR repertoire diversity and expression of genes associated with lymphocyte chemotaxis and activation in MCC patients responding to immune checkpoint inhibition of PD-1 and its ligand PD-L1 [78]. In non-responders, terminally differentiated effector T-cells with a constrained TCR repertoire prevailed. Furthermore, sequential analyses of tumor tissue obtained during immunotherapy revealed a more pronounced and diverse clonal expansion of TILs in responders, indicating an impaired proliferative capacity among TILs of non-responders following checkpoint inhibition [78].

## 5. Chemokines and Cytokines in T-Cell Recruitment

Positive and negative signals for immune cell recruitment play influential roles in regulating the TME [79]. Various families of cytokines regulate tumorigenesis-related processes, including those related to tumor cell survival, senescence, angiogenesis, metastasis, and immune escape [80]. In the early stages of tumor development, chemokines are secreted by both tumor and immune cells, initiating the recruitment of naïve Tregs to the TME [81]. These chemokines can act in an autocrine manner to promote the proliferation and migration of tumor cells to the tumor. In melanoma, specific chemokines and chemokine receptors, such as chemokine (C-C motif) ligand (CCL)27-CCR10, chemokine (C-X-C motif) ligand (CXCL)12-CXCR4, CCL20-CCR6, CCL19/CCL21-CCR7, CCL9/10/11-CXCR3, CCL17/22-CCR4, and sphingosine-1-phosphate (S1P)-S1PR1, are involved in Treg-mediated immune homeostasis [82,83,84,85,86,87]. However, the pathways by which specific chemokines contribute to T-cell homing in MCC are still underexplored.

Currently, there is evidence for the increased expression of CCL17 in virus-positive cell lines compared to virus-negative cell lines [88], suggesting that CCL17 may contribute to the host immune responses targeting virus^+^ MCC. CCL17 is a chemoattractant that helps in the recruitment of CD4^+^ Tregs, T_H_2, and T_H_17 cells, emphasizing its important role in orchestrating the immune responses within the TME [88,89]. A key concept in the field of oncology recognizes tumor-promoting inflammation as a hallmark of cancer. Importantly, oncogenic cellular changes can promote the induction of inflammatory pathways in both pre-malignant and malignant cells [79,90]. Subsequently, this leads to increased expression of various inflammatory mediators (cytokines and chemokines) that further recruit inflammatory cells. These mediators can alter the TME and are therefore targets to prevent the progression and metastasis of disease [79,90].

The release of proinflammatory cytokines into the TME is critical for MCC development. For example, recent work has focused on IL-33, a ligand for ST2 and IL1RAcP (suppression of tumorigenicity 2 receptor and IL-1 receptor accessory protein, respectively). IL-33 is an endogenous danger signal released following cellular damage and is known to activate various immune cells [91,92]. In MCC, IL-33 has been linked with increased risk of malignancy and worse disease prognosis [92,93,94,95,96]. Additionally, its receptor, ST2, has been implicated as a potential checkpoint target, as high ST2 expression has been correlated with low CD8^+^ T-cell cytotoxicity, as well as increased tumor number and size in colorectal cancer [97,98]. In a recent study, MCPyV-derived T antigens were found to induce the IL-33/ST2 axis and potentially serve as a key regulator in the tumorigenesis of virus-positive MCC [93]. This provides evidence that an interplay exists between cytokine expression and MCPyV replication. Further studies are required to identify additional cytokines/chemokines pathways that may affect anti-tumor response.

## 6. Formation of Tertiary Lymphoid Structures

Tertiary lymphoid structures (TLS) are organized clusters of immune cells in non-lymphoid tissues that, unlike secondary lymphoid organs, are formed in response to inflammatory conditions, including within the tumor site [99]. TLS are sites rich in T-cells, with densities that correlate with CD8^+^ T-cells and CD4^+^ T-cells in tumors [100]. Increasingly, TLSs have been suggested to induce long-lasting antitumor responses, with increasing evidence that the identification of TLSs in tumors can be associated with better prognosis and immunotherapy treatment outcomes [101]. Behr et al. first showcased that the presence of TLS significantly correlated with recurrence-free survival of MCC after examining virus-positive cases [102]. Additionally, they observed that TLS were significantly associated with higher CD8:CD4 T-cell ratio in the tumor periphery, but not within the tumor center [102]. Recently, Nakamura et al. examined both virus-positive and negative cases and determined that there are five chemokine genes—*CCL5*, *CCR2*, *CCR7*, *CXCL9*, and *CXCL13*—with significantly higher expressions in TLS-positive samples compared to TLS-negative samples [103]. Regardless of the expression of TLS in MCPyV-positive tissue samples, virus-positive tissue samples had a similar prognosis. On the other hand, virus-negative tissue samples with TLS-positive expression had a similar prognosis to virus-positive tissue samples, while virus-negative/TLS-negative samples had a significantly worse prognosis [101]. Interestingly, the chemokine genes *CXCL10* and *CX3CR1* had different expression levels in the presence of MCPyV infection (compared to lack of infection) in TLS-negative samples. It was also shown that patients with high *CXCL13* or *CCL5* expression had better prognosis compared to those with lower expression, suggesting that chemokine expression profiles may offer insights into the TME and thereby may represent a potential prognostic marker [103].

## 7. MCPyV-Positive and MCPyV-Negative MCC

MCC is categorized as either MCPyV-positive or MCPyV-negative, depending on whether there is a persistent expression of the viral products LT and sT antigens. Both categories respond similarly to anti-PDL-1 therapy, but there is evidence that virus-positive and virus-negative MCC have distinct morphological and immunohistochemical features [104,105]. It currently remains unclear whether virus-positive and virus-negative MCC are two subtypes of MCC or represent two different tumors that share similar features. Therefore, to administer a more targeted treatment approach for MCC, there is a need to explore the specific immune responses and tumorigenic pathways related to these two MCC types.

### 7.1. T-Cell Responses to Oncogenic Merkel Cell Polyomavirus Proteins

Several lines of evidence are compatible with a role for the adaptive arm of the immune system in recognizing and eliciting protective responses to oncogenic MCPyV proteins. Human leukocyte antigen (HLA) class I tetramers have been used to determine CD8^+^ T-cells that recognize unique epitopes of the persistently expressed T antigens of MCPyV [25,42,106]. Despite the induction of T-cells that recognize immunogenic MCPyV capsid proteins and oncoproteins, the MCC tumors continue to progress, suggesting that MCPyV may have unique capabilities that allow them to escape host the immune defenses [25]. Specifically, it has been observed that MCPyV can consistently downregulate Toll-like receptor 9, as well as gene expression associated with NF-κB signaling pathways eventually leading to immune escape.

The diversity or functional avidity of the TCR repertoire that recognizes CD8^+^ T-cell specific tumor epitopes may be associated with the effectiveness of T-cell responses to MCC [106]. Miller et al. explored how primary CD8^+^ T-cells recognize the epitope KLLEIAPNC, which is derived from the MCPyV common T-Ag and has been associated with intra-tumoral infiltration and positive patient outcomes [106]. The authors observed that patients with greater KLL-specific clonotype diversity in their tumors had significant improvements in MCC and recurrence-free survival. Additionally, T-cell clones generally expressing a narrow range of functional avidities were similar within each patient and tumor cell recognition by these clones had a higher functional avidity [106].

### 7.2. T-Cell Responses in Virus-Negative MCC

Several studies have demonstrated that T-cells reactive to MCPyV are present in MCC patients’ blood and tumors [25,42,106,107]. When compared to virus-positive tumors, virus-negative MCC have lower intra-tumoral T lymphocyte counts, including fewer CD3^+^, CD8^+^, CD16^+^, FoxP3^+^, and CD68^+^ cells [108]. However, studies suggest that virus-negative MCCs still exhibit T-cell tumor specificity and that increased lymphocyte infiltration in these tumors is associated with fewer metastases and improved overall survival [25,108]. Additionally, UVR can promote the recruitment of Tregs and the secretion of IL-10, an immunosuppressive cytokine that facilitates immunologic tolerance. This mechanism of Tregs recruitment and IL-10 secretion is a counterproductive process that sustains MCC tumor development and may be an important treatment consideration for virus-negative MCC [109]. Still, the research on the role of T-cell responses in virus-negative MCC remains limited. Future work should aim to elucidate the role of T-cell mediated immunity in the virus-negative subset of MCC.

## 8. Tumor Immune Escape and Targets for Immunotherapy

According to National Comprehensive Cancer Network guidelines, the current immunotherapy treatment paradigm for MCC involves the use of immune checkpoint inhibitors targeting PD-1 (e.g., pembrolizumab, nivolumab) and PD-L1 (e.g., avelumab) for distant metastatic disease [110]. This is due to MCC promoting immune escape by upregulating PD-1 in TILs, resulting in decreased immune function in response to intra-tumoral expression of PD-L1 [111]. Yet, the majority of patients still does not achieve long-lasting clinical responses with these therapies [112]. Combining novel treatment regimens such as checkpoint inhibitors and contemporary therapies that target immune pathways of T-cells may hold a promise in treatment-refractory patients.

Tumor immune escape refers to the critical event in oncogenesis in which selection pressure from the TME results in the emergence of cells that evade immune surveillance and destruction [113]. The mechanisms of immune escape in MCC may involve tumor cells acquiring new features to become either “less visible” or “more resistant” to tumor-killing immune cells [114]. For example, the MCPyV associated with MCC may facilitate the loss of tumor antigen expression by downregulating MHC-I subsequently resulting in impaired presentation of MCC intracellular peptides to CD8^+^ T lymphocytes, hence rendering tumor cells “less visible” [114,115]. On the other hand, tumor cells may become “more resistant” through MCC oncoproteins targeting tumor suppressor pathways such as the p53 pathway [116,117]. As MCC tends to occur in elderly patients, who typically have weaker immune systems as a direct result of aging (i.e., immunosenescence), immunosuppression leading to T-cell dysfunction may represent another mechanism of tumor immune escape [114].

MCC tumor immune escape secondary to impaired T-cell immunity may represent a promising therapeutic target. For example, the natural killer group 2D (NKG2D) transmembrane receptor, which signals cellular stress to the immune system, and whose ligands include the MHC class I chain-related proteins (MIC) A and B, may be evaded by MCC as a means of immune escape [112]. Cancer cells and viruses interfere with MICA- and MICB-induced NKG2D signaling through various mechanisms, including epigenetic silencing via chromatin remodeling [118]. Ritter et al. described how a reversal of MIC A and B silencing ameliorates the immune recognition of MCC [112]. Furthermore, the introduction of histone deacetylase inhibitors to MCC cells both in vitro and in vivo reversed the epigenetic silencing of MICA and MICB, resulting in increased susceptibility of MCC cells to immune-mediated tumor lysis [112,119]. In a follow-up study, they demonstrated that the pharmacological inhibition of histone deacetylases restored HLA class-1 expression [120]. Consequently, this targeted restorative approach may be an effective means to improve viral antigen processing and surface presentation in most MCCs [120].

TCR-based immunotherapies aimed at strengthening the endogenous T-cell response may also augment patient response to PD-1/PD-L1 axis blockade. Chimeric antigen receptor (CAR) T-cell therapy, which involves genetically altering a patient’s T-cells with viral vectors to express CAR constructs, expanding these newly engineered T-cells ex vivo, and reinfusing them into the patient so that they exert anti-tumor effects, may show promise in MCC patients refractory to PD-1/PD-L1 axis blockade [121]. One future approach might involve the infusion of genetically modified CAR T-cells against MCPyV antigens. Similarly, Gavvovidis et al. targeted the MCPyV itself by targeting MCPyV-encoded T antigens using the ABabDII mouse model known to express a diverse human TCR repertoire [122]. These researchers identified naturally processed epitopes of MCPyV-encoded T antigens and isolated TCRs specific to these T antigens. Then, they engineered human T-cells to express these TCRs in vitro and these T-cells were able to recognize MCC tumor cell lines and demonstrate cytotoxic activity [123]. Therefore, isolating TCRs for MCPyV-encoded T antigens may have potential applications in TCR gene therapy designed to augment patient response to current first-line immunotherapies for MCC.

Moving forward, it will be important to elucidate mechanisms explaining the attenuation of CAR T-cell therapy efficacy in MCC. One such mechanism involves effector trogocytosis between tumor-specific cytotoxic T-cells and tumor cells. In this process, cytotoxic T-cells ingest a piece of cancer cell membrane and begin expressing that antigen on their own cell surface, therefore appearing as a cancer cell to other T-cells [124]. Lu et al. demonstrated that tumor-derived factors induce trogocytosis by altering the lipid profile of cytotoxic T-cells, including the depletion of 25-hydroxycholesterol (25HC) by inducing the ATF3 transcription factor to suppress the expression of the 25HC-regulating gene cholesterol 25-hydroxylase (CH25H). The effector trogocytosis was found to weaken anti-tumor immunity, stimulate tumor growth, and impede the efficacy of CAR T-cell therapy [125]. Moreover, the effects of the ATF3-CH25H axis could be reversed using CH25H-armored CAR constructs, suggesting that inhibiting trogocytosis via this pathway may improve the anti-tumor effects of CAR T-cell therapy.

## 9. Potential Role of Extracellular Traps in MCC

Since the discovery of neutrophil extracellular traps (NETs) two decades ago, significant progress has been achieved in defining their role in antimicrobial defense [126]. Extracellular trap (ET) formation is not unique to neutrophils as ETs have been described in other immune cells [127,128,129,130,131,132,133,134,135,136,137]. The work from our lab and others has shown that antigen-specific T-cells can release T-cell ETs that trap and kill microbes [128,138]. These observations suggest a role for ETs in immunity. Additionally, ETs have been shown to have the capacity to either inhibit tumor metastases/development or drive tumor progression [139]. In animal models, the experimental data suggest a pro-tumor effect of NETs on cancer, though these findings have been inconsistent and controversial [139,140]. Currently, the complete role of ETs in MCC is unknown and likely to emerge as other unknown circumstantial factors may also be at play.

### 9.1. Extracellular Traps: Strands in the Dark Web

Studies suggest a pro-cancer effect of ET components [139]. The DNA component of ETs (ET-DNA) has been associated with cancer metastasis in animal models, suggesting that NETs can act as chemotactic factors that attract cancer cells to form distant metastases [141,142]. Other studies in humans have observed an abundance of NETs in the liver metastases of patients with breast and colon cancers and suggested that serum-NET levels could act as predictors of occurrence of liver metastases in early stage breast cancer [143,144,145,146]. Yang et al. identified the transmembrane protein CCDC25 as an NET-DNA receptor on cancer cells that senses extracellular DNA, thus leading to the activation of the ILK-β-parvin pathway enhancing cell motility [146]. In all, these studies lay the groundwork for the investigation of how ETs might be initiated in and directed to target MCC or particular antitumor responses.

### 9.2. Extracellular Traps: Keeping the Peace

More than 90% of the MCC patients demonstrate normal immune function but fail to clear tumors [114,147]. However, the presence of CD8^+^ T-cells in MCC tumors is positively associated with patient survival [55,56,108,148]. Still, MCPyV-positive MCCs have been hypothesized to escape immunological elimination by infiltrating T-cells via repressing their activation, although the specific mechanism of immune evasion remains unknown. It also remains unclear whether therapies stimulating intra-tumoral T-cell recruitment and activation, and ET release, can be successful as an immunotherapeutic strategy. Some studies have suggested that baseline levels of ETs can produce an anti-tumor effect, thereby killing cancer cells or limiting tumor growth and metastasis through immune activation within the TME [139]. Importantly, ETs derived from neutrophils suppress the proliferation of cancer-associated intestinal microbial populations, hence inhibiting the proliferation and metastasis of colorectal cancer cells [140].

Altogether, these studies challenge our current understanding of the ET-MCC tumor microenvironment. Promising future areas of investigation include studying 1) how ETs produced by the same cells within the TME simultaneously inhibit the proliferation of tumor cells and enhance their motility, 2) whether ETs derived from different immune cells are similar in composition, and 3) how ETs are initiated and regulated within the TME. Improved understanding of the role of ETs in cancer will facilitate the answering of these and other relevant questions. Above all, targeting NET-dependent metastases may be an appealing therapeutic strategy for preventing cancer metastasis, including in MCC [146].

Some studies have reported ETs to be composed of mitochondrial DNA (mtDNA) suggesting that the mitochondria can be involved in the process of ET formation and inflammatory response [149]. Tumors that cause the disruption of mitochondrial homeostasis can lead to the accumulation of mtDNA, a dominant driver of systemic inflammatory response. mtDNA is recognized as an “alarmin” that stimulates immune cells at physiological concentrations [150,151,152,153,154] Both mtDNA and genomic DNA released from dying cells activate the cyclic GMP-AMP synthase (cGAS) stimulator of the interferon genes (STING) pathway and type I IFN signaling. Since cancer cells often maintain high levels of damaged DNA, this could potentiate the cGAS–STING pathway and the subsequent secretion of type I IFNs as well as pro-inflammatory cytokines [155,156,157,158,159]. These molecules can not only display direct cytotoxic antitumor activity, but also promote tumor antigen-specific intra-tumoral T-cell infiltration, activation, and tumor repression [156,157,158,159,160,161]. Further studies investigating the critical role of the ET:cGAS–STING connection in mediating antitumor T-cell responses are therefore warranted.

## 10. Novel Methods for MCC Studies

With the introduction of novel technologies in recent years, exciting opportunities for hypothesis testing in MCC research and the more precise characterization of the phenotype and functional status of immune cells are possible. For example, FAUST, a new machine-learning method for single-cell cytometry studies, offers an unbiased method for cell population discovery and annotation [162]. Already, it has been used to test cytometry data generated from blood isolated from patients with MCC who received pembrolizumab as part of a phase 2 clinical trial [162]. FAUST was able to visualize baseline T-cells in the blood of subjects (pre-treatment) and associate these samples with the subjects’ responses to therapy in the clinical trial. Additionally, the development of a viable mouse model that can test the role of MCPyV T-Ags in MCC development has recently emerged, providing a new model for addressing scientific questions related to MCC. Verhaegen et al. have generated and validated transgenic mouse strains with the doxycycline-inducible expression of MCPyV sT and LT antigens that carry internal ribosome entry site-driven (IRES-driven) red and green fluorescent protein reporters [163]. This development offers future areas to explore and may be helpful for future research seeking to examine the MCC biological mechanisms that underlie viral T-Ag-driven tumorigenesis, as well as the testing of novel therapeutics.

## 11. Conclusions

Since MCC was first described in 1972 [164], substantial research efforts have been dedicated toward the characterization of the cancer, as well as its optimal management and treatment. Namely, uncovering the pathophysiology of MCC and its relationship with the host immune response has been of interest. Our current understanding suggests that TILs play diverse roles during MCC development with observations that the increased abundance of CD8^+^ T-cells is associated with favorable outcomes, while other T lymphocyte subsets, such as Tregs, contribute to an immunosuppressant tumor microenvironment that can promote tumor escape [38]. Still, unanswered questions related to T-cell responses in MCC remain and must be addressed.

Additional research is needed to identify not only the relevant T lymphocyte populations, but also the specific cytokines they secrete, which can modulate the TME and alter immune surveillance and response. Currently, it is known that cytokines and chemokines contribute to the inflammation and recruitment of cells to the tumor site and that the dysregulation of their balance can impair immune responses to control tumor growth [81,165]. For example, pro-inflammatory cytokines, such as TNF-α, IL-17, IL-22, and IL-23 have been associated with tumorigenesis, whereas IFN-γ and IL-12 have been associated with mediating cytolytic attack [81]. On the other hand, IL-10 and TGF-β are immune modulating cytokines that contribute to immunosuppressive TME and promote cancer progression and metastasis [166]. Additionally, chemokines are critical for the recruitment of immune cells and can either promote homeostasis or drive inflammation. However, the categorization of these various regulatory and inflammatory elements as prognostic factors for MCC-related outcomes can be difficult as their roles are typically context dependent and are related to the balance of all immune players in the TME. Therefore, it will be of interest to understand the immune networks that are driven by both MCPyV and UV radiation within the TME. It is important to highlight that the tumor origin cell in MCC has not yet been identified, so further studies aiming to characterize the origin cell may lead to the identification of an alternative target in therapy for MCC.

The clarification of the unique differences of the two etiologies of MCC, specifically the T-cell responses of virus-negative MCC, for which few studies detail the immune responses involved, can inform the future work on targeted therapies. While we understand that both etiologies contribute to similar phenotypical characteristics of the tumor, the elucidation of the unique and common pathways shared between the two etiologies as well as the interplay of genetic, epigenetic, and environmental factors involved in MCC are important for more targeted therapies. Given the well-understood role of transplantation in skin cancer development, additional effort to understand how specific types of immunosuppressants may differentially affect the distribution of T lymphocytes and their functionality within the TME may be vital. The knowledge of how to create therapies that boost antitumor responses of TILs seems promising. As we continue to define the molecular fingerprint of TILs, the identification of new targets for immunotherapy is imminent.

## Figures and Tables

**Figure 1 cancers-14-06058-f001:**
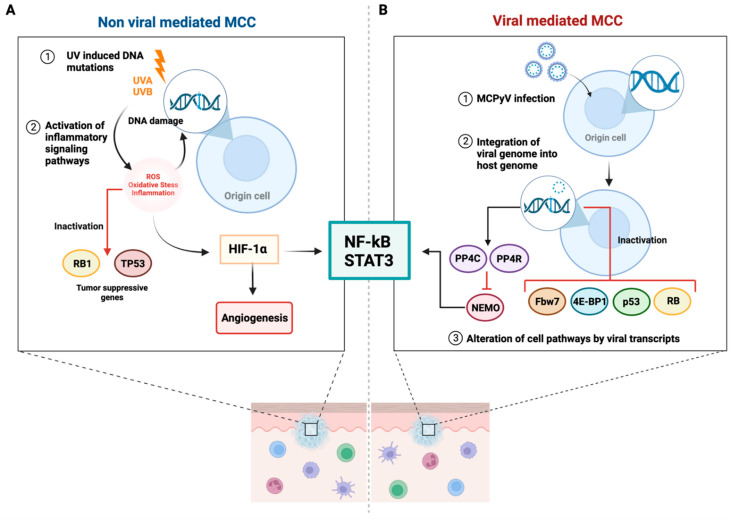
Proposed etiological pathways for Merkel cell carcinoma (MCC). (**A**) UVA and UVB rays cause DNA damage, which generates ROS that contributes to the oxidation of DNA bases. This results in the accumulation of mutations, especially in the TP53 and RB1 genes, leading to the inactivation of tumor suppressive pathways. Additionally, increased ROS levels increase HIF-1α activity. HIF-1α activates NF-κB/STAT 3 pathways and angiogenesis, leading to increased tumor cell proliferation and the promotion of metastasis. (**B**) 80% of MCC cases are attributed to MCPyV etiology, which is first initiated by MCPyV infection. Following integration of the viral genome into the host genome, two neoantigens: sT and LT antigens, are expressed. sT and LT are involved in the inhibition of cell pathways: retinoblastoma, tumor suppressor p53, 4E-BP1, and Fbw7, which contribute to increased cell cycle progression, decreased tumor suppression, increased translation, and decreased proteasomal activity, respectively. Activation of PP4C and PP4R1 subunits inhibit NEMO, which decreases antiviral response and leads to failure of NF–kB signaling. Abbreviations: UVA, Ultraviolet A; UVB, Ultraviolet B; ROS, reactive oxygen species; RB, retinoblastoma; TP53, Tumor protein P53; 4E-BP1, eukaryotic translation initiation factor 4E-binding protein 1; HIF-1α, Hypoxia-inducible factor 1-alpha; STAT3, Signal transducer and activator of transcription 3; MCPyV, Merkel cell polyomavirus; sT, small tumor; LT, large tumor; PP4C, protein phosphatase 4 catalytic, PP4R1, protein phosphatase 4 regulatory; NEMO, NF-kappa-B essential modulator; NF-κB, Nuclear factor-κB; Fbw7, F-Box and WD Repeat Domain Containing 7. Created with Biorender.com.

**Figure 2 cancers-14-06058-f002:**
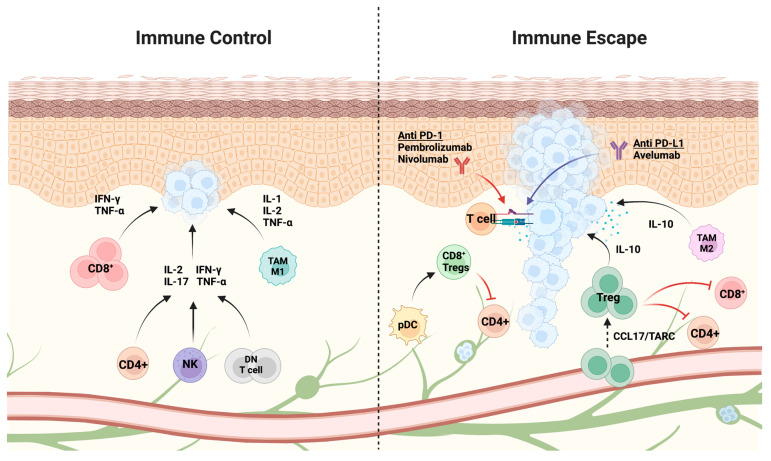
Immune cells and secretory molecules within the tumor microenvironment (TME) shape the course of Merkel cell carcinoma. Shown here are several key cellular components and secretory molecules that mediate the balance of cytotoxic versus suppressive responses surrounding tumor cells. Presence of CD8^+^ T-cells, CD4^+^ T_H_1, NK cells, DN γδ T-cells, and TAMs contribute to antitumor effects through the secretion of inflammatory cytokines. During early stages of tumor development, immune cells, namely NK and CD8^+^ T-cells, recognize and eliminate immunogenic cancer cells, promoting the selection of less immunogenic cancer cells. Activated M1 macrophages contribute to antitumor responses, but as the tumor progresses, shift toward M2-like-polarization of TAMs contributing to anti-inflammatory pro-tumorigenic responses. Selection of tumor cells that evade or resist tumor killing cells lead to development of the tumor. Tumor cells can secrete molecules that recruit suppressive cells, including Tregs, which can suppress CD4^+^ and CD8^+^ T-cell proliferation and activity. The TME can activate pDCs to secrete cytokines that induce the generation of CD8^+^ Tregs, leading to inhibition of CD4^+^ T-cells. Binding of PD-1 receptor on T-cells to PD-L1 expressed on tumor cells results in the suppression of T-cell proliferation and the host immune response. Antibody blockade of PD-1 and PD-L1 disrupts this tumor escape mechanism, paving the way for development of new therapeutics for MCC. Abbreviations: NK, Natural Killer; T_H_1: Type 1 T helper; DN, double negative; TAM, tumor associated macrophages; Tregs, Regulatory T-cells; pDCs, plasmacytoid dendritic cells; PD-1; programmed cell death protein 1 receptor; and PD-L1, programmed death ligand 1. Adapted from “Skin Epithelium Cross Section with Blood and Lymphatic Vessels (Layout)”, by Biorender.com (2022). Retrieved from https://app.biorender.com/biorender-templates accessed on 10 November 2022.

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
