# Peer review of "T-Cell Mediated Immunity in Merkel Cell Carcinoma"

_cancers, 2022, doi:10.3390/cancers14246058_

Round 1
Reviewer 1 Report
This is an extremely well-written and interesting, comprehensive literature review on Merkel cell carcinoma. The authors succeed in digesting most, if not all of the key literature and do a great job in organizing the information in their text and figures. I commend the authors for this exemplary work.
Major
1. Lines 109-111: please consider commenting on T-cell exhaustion here, as this is another parameter affecting the host's immune response. This phenomenon is already discussed later on within this manuscript and appropriate citations have been used.
2. Lines 261-264: Please consider including more contemporary references and discussing them appropriately. The current reference is 20 years old and thus, by no means should be considered reflective of our current understanding of the field. Examples 10.1158/2159-8290.CD-21-1059 and 10.1038/nrclinonc.2015.105
3. Line 418: Sections 9.2 and 9.3 appear to have a distinct, more literature-like style. While this is totally appropriate for a lit review, in this particular case it stands out as a heterogeneous part of the manuscript. Authors are encouraged to homogenize these sections so that they are in line with the style of the rest of the manuscript.
Minor
1. Lines 63-64: Please revise to Feng et al.- Usually it is advised to mention the first author, and not the lab he belongs when referencing to a published work.
2. Line 79: Consider replacing virus with MCPyV
3. Line 298: Per the authors comment earlier, there is no demonstrated definitive causality between MCPyV and MCC. I would encourage the authors to avoid the term "viral-mediated" because it implies direct causal association. Alternatively elaborate more on the causal relationship.
4. Line 348: "The majority of patients still does not"
5. Line 416: The complete role of ETs in MCC is unknown.
Reviewer 2 Report
Review - cancers-2058554 - T-cell mediated immunity in Merkel cell carcinoma
Kelsey Ouyang , David X. Zheng , George W. Agak *
This is an interesting and well written review focused on an interesting topic. It comprehensively describes the T-cell mediated immunity in Merkel cell carcinoma, which is a rare non-melanoma skin tumor. It particularly describes and discusses the impact of tumor infiltrating lymphocytes, mainly the CD4+, CD8+, and regulatory T-cell (Tregs) responses during MCC initiation and progression. Clinical application of immunotherapies is also described. The work is well done and gives important information on the role of T-cell mediated immunity in the onset of this skin tumor. It is an informative and well written review. A large variety of references in the field have been included. The ms is suitable for publication from my point of view. However, it requires some minor corrections:
I have some minor comments/observations aimed in improving the work
1. The work should be carefully revised for the presence of several minor typos/grammar errors
2. Line 31 The estimated 5-year survival rate is 63% according to the SEER - Surveillance, Epidemiology, and End Results Program (https://seer.cancer.gov/)
3. Line 27 Please remove the green annotations
4. In figure 1 I suggest including the important implication of MCPyV oncoproteins LT and sT in tumor initiation, given that both viral proteins play a key role in viral-driven MCC onset and progression are also mentioned in the figure caption
5. Lines 135-141 these sentences seem to be in gray
6. Line 169 LT does not present “capsid-derived antigens”, which are referred to viral capsid proteins such as VPs, i.e., VP1 and/or VP2
7. Lines 183 these studies evaluated the anti-viral immunologiacakl response against Viral-like capsid proteins, which is high in terms of seroprevalence in healthy subjects (https://journals.asm.org/doi/10.1128/JCM.01691-09), while circulating IgG antibodies against LT and/or sT in the healthy population are almost completely absent
8. General observation, please revise the text for including the complete name of all proteins/factors which were mentioned as acronyms only. An example can be line 194 for IL-10, that should be “interleukin-10 (IL-10)”.
9. Line 256 MCC cell lines?
10. Line 270 the meaning of ST2 should be explained
11. Lines 289-292 MCC tissues or cell lines?
12. Line 300 this annotation “large T (LT) and small T (sT) antigens” should has been used the first time both proteins were mentioned in the text
13. Line 371-376 relevant information on this important topic has recently been described in detail in this review from frontiers in oncology (PMID: 35350569). I believe this reference should be included. Moreover, this additional work from Ritter et al reporting the epigenetic priming of HLA Class-I Antigen Processing Machinery should also been considered (doi: 10.1038/s41598-017-02608-0)
14. Line 435 “the mechanism…is still”.
15. Line 495 “it is known that..”
16. Line 516 I suggest including “epigenetic”
